# An Amidase Contributes to Full Virulence of *Sclerotinia sclerotiorum*

**DOI:** 10.3390/ijms231911207

**Published:** 2022-09-23

**Authors:** Wei Li, Junxing Lu, Chenghuizi Yang, Kate Arildsen, Xin Li, Shitou Xia

**Affiliations:** 1Hunan Provincial Key Laboratory of Phytohormones and Growth Development, College of Bioscience and Biotechnology, Hunan Agricultural University, Changsha 410128, China; 2Michael Smith Laboratories, University of British Columbia, Vancouver, BC V6T 1Z4, Canada; 3Department of Botany, University of British Columbia, Vancouver, BC V6T 1Z4, Canada; 4College of Life Science, Chongqing Normal University, Chongqing 401331, China

**Keywords:** *Sclerotinia sclerotiorum*, oxalic acid, amidase, virulence

## Abstract

*Sclerotinia sclerotiorum* is one of the most notorious and ubiquitous soilborne plant pathogens, causing serious economic losses to a large number of hosts worldwide. Although virulence factors have been identified in this filamentous fungus, including various cell-wall-degrading enzymes, toxins, oxalic acids and effectors, our understanding of its virulence strategies is far from complete. To explore novel factors contributing to disease, a new pipeline combining forward genetic screening and next-generation sequencing was utilized in this study. Analysis of a hypovirulent mutant revealed that a mutation in an amidase-encoding gene, *Sscle_10g079050*, resulted in reduced virulence. This is a first report on the contribution of an amidase to fungal virulence, likely through affecting oxalic acid homeostasis.

## 1. Introduction

*Sclerotinia sclerotiorum* (Lib.) de Bary is an economically devastating fungal pathogen with an extremely broad host range. Infection is initiated through multiple orchestrated strategies that ultimately incite rapid host tissue maceration, causing diseases in over 600 plant species [1]. Largely through reverse genetic analysis, genes involved in apothecia and ascospore development [2,3,4], mycelial growth [5,6], appressorium establishment [7,8], sclerotial formation and development [9,10,11], and fungal virulence [7,12,13] have been identified over the last decades. Many virulence factors were confirmed to play critical roles in host tissue disruption and degradation, such as cell-wall-degrading enzymes (CWDEs) and toxins [14]. In addition, some small secreted proteins functioning in both host defense suppression and inducing host cell death have been identified [15,16].

The full virulence of *S. sclerotiorum* requires controlled oxalic acid (OA) accumulation. OA levels are regulated through its biosynthesis and degradation [7,12,13]. Defective OA synthesis leads to severely compromised virulence, implicating the importance of OA in *Sclerotinia* pathogenesis. Proposed roles of OA include creating a low pH environment available for hydrolytic enzyme activities [17,18,19], contributing to calcium chelation and cell wall breakdown, reducing calcium exposure toxicity during invasion of growing hyphae [20,21], eliciting apoptotic programmed cell death to allow fungal pathogenicity [22], and disruption of chloroplast function [23]. Moreover, OA suppresses and promotes host reactive oxygen species (ROS) at different stages of infection to achieve successful disease development [22,24,25].

With the increasing availability of sequenced whole genomes of fungal pathogens, reverse genetic analysis for genes of interest are available for functional investigation. In terms of novel gene discovery, it is inevitably limited by bias. Recently, Xu et. al. [26] provided an effective pipeline, which combines forward genetic screening with high-throughput next-generation sequencing (NGS) to enable quick gene discovery and confirmed the reliability of this method in *S. sclerotiorum*.

Here, we performed a forward genetic screen for virulence-related genes using the same mutagenized population acquired from Xu et. al. [26]. Mutants defective in virulence were isolated and NGS was used to identify the candidate genes. Specifically, we describe a recovered mutant with a mutation in an amidase-encoding gene. Knockout and transgene complementation experiments confirmed that this amidase mutation is responsible for the observed virulence phenotype. Thus, our study uncovered an important role amidase plays in fungal virulence, probably by affecting OA homeostasis.

## 2. Results

### 2.1. Screening for S. sclerotiorum Mutants with Reduced Virulence

Using a UV-mutagenesis based forward genetics pipeline [26], we generated a mutagenized population originated from single haploid ascospores. Our initial screen aimed for mutants deficient in sclerotial formation and melanization. The same population was then grown to search for mutants which exhibit reduced virulence on lettuce leaves. Here, we report on the characterization of mutant 5–10. On lettuce leaves, wild-type (WT) *S. sclerotiorum* caused large lesions 48 h post-inoculation, while 5–10 mutant formed smaller ones (Figure 1A,B). To test whether the reduced virulence was associated with hyphal development, we examined the growth rate of the mutant. As shown in Figure 1C,D, mycelial growth of the 5–10 mutant is comparable with WT. Accordingly, the defects in 5–10 seems specific to fungal virulence.

### 2.2. Less Oxalic Acid Is Observed in 5–10 Mutant

As oxalic acid (OA) is a key virulence factor in *S. sclerotiorum* [20], we examined whether the level of OA in 5–10 is altered. When OA was measured both qualitatively and quantitatively (Figure 2A,B), 5–10 indeed showed significantly reduced OA amounts. Thus, the reduced virulence observed in 5–10 is likely due to its reduced OA levels during mycelial growth.

### 2.3. Mutation in Sscle_10g079050 Is Responsible for the Mutant Phenotypes of 5–10

To identify the mutation responsible for the observed phenotypes of 5–10, the full genome of 5–10 was subjected to whole genome next-generation sequencing (NGS). The genomic sequence of an unrelated mutant *pnk1* was used as negative control to exclude background mutations that occured in both 5–10 and *pnk1* [26]. From whole genome sequence comparisons, a total of 77 SNPs (single nucleotide polymorphisms) and 689 INDELs (insertions and deletions) were captured in 5–10, including mutations in exonic, intronic, upstream, downstream, and intergenic regions. After manual elimination of synonymous, intronic, intergenic, and non-homozygous mutations, 2 exonic SNPs remain that led to nonsynonymous changes in the 5–10 genome. These two SNPs in *S**scle_06g054440* (L1407S) and *S**scle_10g079050* (L82S) became the prime candidate mutations for 5–10.

To determine which of the two mutations is responsible for the 5–10 phenotypes, transgenic complementation of *Sscle_06g054440* and *Sscle_10g079050* was individually conducted (Figure 3 and Figure 4). Only when *S**scle_10g079050* was transformed into 5–10, did the transgenic lines yield similar level of acidic substances and display almost identical virulence compared to WT (Figure 3B–E). Transformation of the other candidate gene had no effect on virulence (Figure 4A,B). Therefore, we conclude that the mutation in *Sscle_10g079050* is likely responsible for the 5–10 phenotypes.

To investigate the function of *Sscle_10g079050* (protein accession number, APA13135.1/XP001584936.1), homology alignment and phylogenetic analysis were conducted within *S. sclerotiorum* (Figure 5A). This gene appears to be a member of a large gene family, and the highest identity with its paralogous protein (*Sscle_04g033530*) was only 42.34%. Phylogenetic analysis among multiple species indicated that the transcripts of *Sscle_10g079050* were conserved in fungi (Figure 5B,C). Based on the annotation of its homologous proteins, *Sscle_10g079050* was predicted to possess a function of amidase or acetamidase. The point mutation in 5–10 caused an amino acid to change from Leucine to Serine (L82S) and this Leu is conserved in fungi, especially in *Leotiomycetes* (Figure 5C), suggesting a critical role of this residue to its function. As *Sscle_10g079050* encodes an enzyme catalyzing the hydrolysis of an unknown amide, it does not seem to be directly involved in OA biosynthesis or catabolism. It likely affects the OA level through indirect means.

### 2.4. 5–10 and Sscle_10g079050 Knockout Mutants Exhibit Similar Hypovirulence

To further confirm that the mutation in *Sscle_10g079050* is responsible for the hypovirulence in 5–10, targeted gene replacement by homologous recombination was performed in WT background. The *Sscle_10g079050* transformants were selected on PDA with 50 μg/mL of hygromycin, and two independent transformants, KO1 and KO2, were obtained (Figure 6A). PCR analysis showed that the WT copy of *Sscle_10g079050* was not fully absent in the knockout strains, suggesting that the mutants generated likely represent partial loss-of-function alleles (Figure 6B,C). The virulence in KO1 and KO2 nevertheless were reduced compared with WT on both lettuce and Arabidopsis (Figure 6D–G). Therefore, we conclude that the mutation in *Sscle_10g079050* is responsible for the hypovirulence in 5–10. As knocking out *Sscle_10g079050* resulted in attenuated virulence, the encoded amidase serves as a positive regulator of virulence. When its gene expression pattern was analyzed from publicly available RNA-Seq datasets [27], we found that it is up-regulated after 24 h of infection (Figure 6H), which supports its critical role in virulence.

## 3. Discussion

Historically, virulence-related genes in *S. sclerotiorum* were generally identified based on either homology with well-characterized genes from other pathogens, or genes encoding secreted proteins with effector functions. However, these approaches have limitations and biases. In this study, we identified an amidase-encoding gene (*Sscle_10g079050*) using an effective and reliable pipeline which combined forward genetic screening with NGS. Mutants of this gene contained less OA and showed reduced virulence. It is the first evidence that an amidase might contribute to the homeostasis of OA in fungi.

Amidases are versatile enzymes owing to their broad substrate specificity [28]. They are widespread in all organisms including bacteria, fungi, plants, and animals [29,30]. According to their primary structure, amidases are divided into three families, namely amidase signature (AS) family, nitrilase superfamily, and FmdA_AmdA family [31]. *Sscle_10g079050* belongs to the AS family, which carry a highly conserved stretch of approximately 130 amino acids known as the AS sequence [32]. The crystal structure of analyzed AS enzymes share an α/β sandwich folded core domain, containing a Gly/Ser-rich motif and Ser-cis Ser-Lys catalytic triad [33,34,35]. After sequence alignment with Pam (PDB ID: 1M21), a structural analyzed AS enzyme determined in *Stenotrophomonas maltophilia*, the catalytic triad of *Sscle_10g079050* was predicted as Ser^202^-Ser^226^-Lys^127^ (Appendix A). According to the 3D structure model, the residue Leu^82^ was located sterically far from that core region but spatially close to Lys^127^(Appendix A). We speculate that the substitution of Leu^82^ might affect the catalytic function in an indirect way, leading to loss of the enzymatic function.

In filamentous fungi, OA is derived from either the hydrolysis of oxaloacetate or the oxidation of glyoxylate or glycolaldehyde [20], and OA biogenesis in *Ascomycota* species relies predominately on oxaloacetate acetylhydrolase (OAH)-mediated oxaloacetate hydrolysis [12]. As a metabolic intermediate of both the tricarboxylic acid (TCA) and glyoxylate cycle, malate or pyruvate might be an oxaloacetate precursor in the OA biosynthesis [36,37,38,39]. In addition, many enzymes involved in lipid metabolism are required for virulence, such as isocitrate lyase in *Magnaporthe grisea*, *Leptosphaeria maculans*, and *Colletotrichum lagenarium* [40,41], and malate synthase in *Stagonospora nodorum* and *S. sclerotiorum* [42,43]. It is well known that isocitrate lyase and malate synthase are key enzymes in glyoxylate cycle, but whether they affect pathogenic processes in an OA-dependent manner remains unclear. Amidases diverged widely with regard to substrate specificity and function. A large spectrum of amide substrates, including aliphatic amides, aromatic amides, and *α*-hydroxyamides were found to be catalyzed by AS enzymes [44,45,46]. One-way *Sscle_10g079050* affects OA accumulation might be through hydrolyzing suitable amide-containing substrates during OA biosynthesis.

Amidases in the AS family usually exhibit both hydrolytic and acyl transfer activities. The intracellular transport of acetyl coenzymeA (acetyl-CoA) is a valuable component of the glyoxylate cycle, and a peroxisomal carnitine acetyl transferase (Pth2) identified in *Magnaporthe grisea* was found to facilitate acetyl-CoA transport specificity, which was required for both *β*-oxidation of fatty acid and pathogenicity [47,48]. With this in mind, it is possible that *Sscle_10g079050* could also play a role in glyoxylate pathway by its acyl transfer activity. Moreover, OA level is also indirectly affected by the redox status of *S. sclerotiorum* [10]. Thus, *Sscle_10g079050* may affect OA accumulation by participating in maintenance of redox environment. The amidase may even affect OA-independent processes involved in virulence.

*Sscle_10g079050* is predicted to be a cytoplasmic enzyme, and it contains a N-terminal cytoplasmic tail (1 to 463aa) and a C-terminal single transmembrane segment, implying that it might anchor to membrane and cannot be accessible to other organelles. Since malate and citrate biosynthesized by the TCA cycle or the glyoxylate cycle can diffuse across the membrane barrier to be exported from organelles and finally be transformed into oxaloacetate, *Sscle_10g079050* may co-locate with malate or citrate in the cytosol pool, and play a role in oxaloacetate or OA metabolism processes. In the future, biochemical and chemical studies will be essential for identifying the exact substrate of this amidase to illuminate its precise mechanism of affecting OA levels and pathogenic processes.

## 4. Materials and Methods

### 4.1. Fungal Strains and Culture Condition

Wild-type strain *S. sclerotiorum* 1980 and mutant were maintained on standard PDA (potato dextrose agar, Bio-Way Technology, Shanghai, China) medium at 20 °C and stored on PDA slants at 4 °C or as sclerotia. Transformants (deletion mutants and complemented strains) were grown on PDA containing 50 μg/mL hygromycin B (Roche).

### 4.2. Plant Infection Assay

Virulence of the strains was tested on detached leaves of lettuce and *Arabidopsis thaliana* (ecotype Col-0). Mycelial agar plugs 2 mm in diameter were inoculated on detached leaves of *A. thaliana* (5 mm in diameter on lettuce) placed on moistened paper towels in Petri dishes (90 mm diameter), with five repeats. Photos were taken at 48 h post-inoculation. The pathogenicity test was repeated three times with similar results. Lesion areas were measured by ImageJ.

### 4.3. Screening for Mutants with Deficiency in Virulence

Ascospores were collected from apothecia of wild-type (WT) strain *S. sclerotiorum* 1980. The UV mutagenesis was conducted on ascospores as previously described [26]. The mutagenized population was initially screened using lettuce leaves obtained from supermarkets. The putative mutants were then confirmed with multiple repeats on both lettuce and Arabidopsis.

### 4.4. Genomic DNA Extraction and NGS

Fresh fungal mycelia were cultured in PDB for one week. Surface floating mycelia were harvested and ground to fine powder in liquid nitrogen. The genomic DNA extraction was conducted by cetyltrimethylammonium bromide (CTAB) method from the mutant and WT strains [49]. The crude extract was further purified for NGS with a commercial service (Novogene, Bioinformatics Technology Co., Ltd., Beijing, China). DNA degradation and contamination were evaluated on a 1% agarose gel. Paired-end library was built by Novogene, using Illumina sequencer NovaSeq 6000, and clean reads were aligned to the *S. sclerotiorum* genome (ASM185786v1).

### 4.5. Candidate Genes Identification

The sequence reads from NGS were mapped to the reference genome of WT strain *S. sclerotiorum* 1980. Mutations were identified by SAMtools with default parameters [50]. Annotation of the mutations was performed with germline short variant discovery (SNPs + INDELs) based on GATK best practices [51]. False mutations in repetitive sequences were manually removed as previously described [52].

### 4.6. Sanger Sequencing of Sscle_10g079050 in 5–10

*Sscle_10g079050* gene fragment was amplified with SeqF and SeqR primers and genomic DNA of 5–10 was isolated by CTAB method and used as template. A 660-bp PCR product was then sequenced using universal primer by Sangon Biotech in China.

### 4.7. Target Gene Knockout and Transgene Complementation

The split marker method was used to generate *Sscle_10g079050* gene replacement constructs [53]. In brief, 1,102-bp upstream and 655-bp downstream flanking fragments of *Sscle_10g079050* were amplified with the primer pairs 5–10-10g KO 1F/2R and 5–10-10g KO 3F/4R, respectively, and were fused with two overlapped *hph* fragments amplified with the primers HYG-F/HY-R and YG-F/HYG-R from pUCH18 plasmid (shared by D. Jiang from Huazhong Agricultural University in China) [54]. All primers used for PCR are listed in Appendix A. Gene flanking fragments were fused with two *hph* fragments, respectively, by homologous recombination methods. The amplified PCR products were co-transformed into protoplasts of WT as described [54]. Transformants were selected on fresh PDA plates supplemented with hygromycin B. The individual germinating transformants were transferred onto new PDA plates with hygromycin B. Fresh mycelia were used to perform colony PCR with a pair of primers, 5–10-10g KO 7F/8R (Appendix A), to confirm the insertion of the hygromycin gene. Positive transformants were inoculated onto new PDA plates with hygromycin B by hyphal tip transfer.

Two additional rounds of the same colony PCR and selection were conducted to accumulate more nuclei with hygromycin insertions. Then, the transformants were cultured in PDB for 2 days. The fresh mycelia mat was used to produce protoplasts as mentioned above. The resulting protoplasts were diluted with PDB supplemented with hygromycin B to 10^3^ protoplasts per milliliter. A 1-mL suspension was evenly spread on PDA plates supplemented with hygromycin B and dried. The germinating colonies were transferred onto individual PDA plates with hygromycin B. Transformant DNA was extracted using the CTAB method, which were later used as template to conduct PCR with four pairs of primers as shown in Figure 6A. The transformants with the target gene being homogeneously knocked out were chosen for further experiments.

For transgenic complementation, a knock-in method was performed to accomplish the in-situ complementation by candidate gene homologous recombination. As described in the knockout protocol, a 3307-bp fragment including upstream and genomic DNA and a 655-bp downstream flanking fragment were PCR-amplified. The resulting two fragments were fused with *hpg* fragments, respectively, and the amplified PCR products were co-transformed into protoplasts of 5–10 as described above. SNP primers (Appendix A) designed on the mutation site were used to check the insertion of *Sscle_10g079050* gene.

### 4.8. OA Analysis

Mycelial plugs (5 mm diameter) with growing hyphal tips were inoculated on PDA medium containing 100 μg/mL bromophenol blue to qualitatively measure oxalic acid. Three mycelia-colonized plugs were also inoculated in PDB medium containing 100 μg/mL bromophenol blue for UV absorbance test after 48 h oscillation with three repeats. OA contents were detected using Oxalic Acid Content Detection Kit following the manufacturer’s instructions (Solarbio Life Science, Beijing, China).

### 4.9. Phylogenetic Analyses

Homologous proteins were searched within *S. sclerotiorum* and among different species. The phylogenetic tree of selected homologous proteins was generated using MEGA7.0 with the maximum likelihood (ML) algorithm. Bootstrap analysis with 1000 replications was performed to assess group support.

### 4.10. Protein Subcellular Localization and 3D Structure Prediction

Protein subcellular localization prediction was conducted utilizing the online tools, WoLF PSORT (https://wolfpsort.hgc.jp/, accessed on 15 November 2021) and GenScript PSORT (https://www.genscript.com/tools/psort, accessed on 10 December 2021). Protein 3D structure model prediction was generated by SWISS-MODEL (https://www.expasy.org/resources/swiss-model, accessed on 27 July 2022).

## Figures and Tables

**Figure 1 ijms-23-11207-f001:**
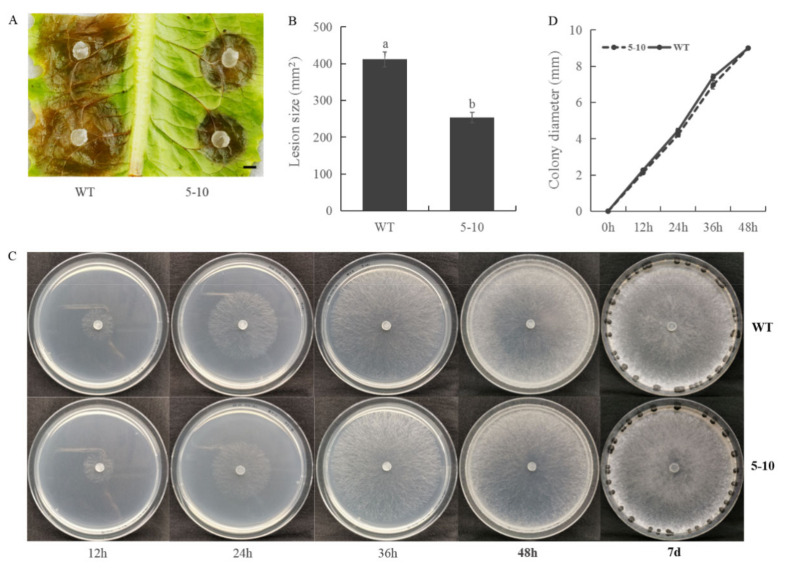
Virulence on lettuce leaves and mycelial growth phenotypes of the 5–10 mutant. (**A**) Necrotic lesions caused by *S. sclerotiorum* infection on lettuce leaves. Bar = 5 mm. Data were recorded at 36 h post-inoculation (hpi). WT: wild-type. (**B**) Quantification of lesion size of WT and 5–10 on lettuce leaves. ImageJ was used to quantify the lesion size. Statistical significance was analyzed using Student’s *t*-test between WT and the 5–10 mutant. a and b represent different significance levels. (**C**) Colonial morphology and mycelial growth of WT and 5–10 strains after 12 h, 24 h, 36 h, 48 h and 7 days on potato dextrose agar (PDA) media. (**D**) Colony diameter of WT and 5–10. Strains were grown on PDA medium and colony diameters were measured every 12 h. All experiments were conducted at least three times with similar results. Error bars represent standard error (SE). Statistical significance was analyzed using Student’s *t*-test between WT and the 5–10 mutant.

**Figure 2 ijms-23-11207-f002:**
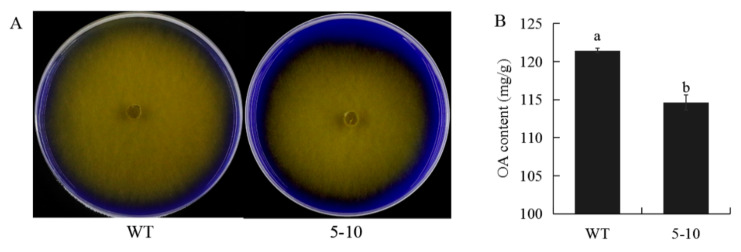
The 5–10 mutant strain produces less oxalic acid (OA). (**A**) Image of mycelia of WT and 5–10 grown on PDA medium containing bromophenol blue. Bar = 10 mm. (**B**) OA content of WT and 5–10 strains. Experiments were conducted at least three times with similar results. Error bars represent SE. Statistical significance was analyzed using Student’s *t*-test between WT and the 5–10 mutant. a and b represent different significance levels.

**Figure 3 ijms-23-11207-f003:**
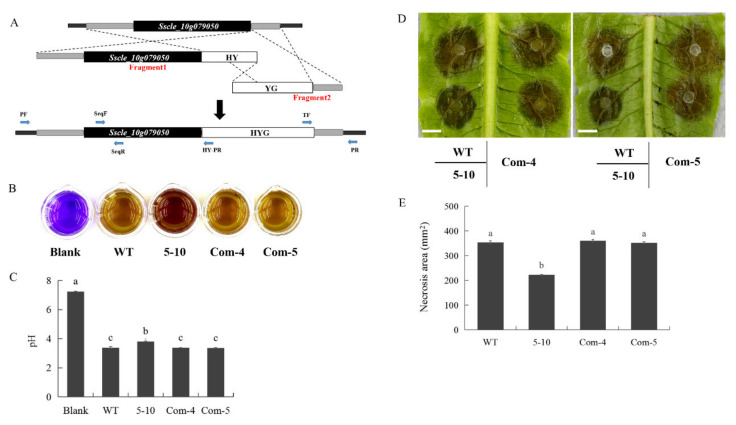
Transgenic complementation of the 5–10 mutant by *Sscle_10g079050*. (**A**) Knock-in method illustration for targeted gene complementation in *S. sclerotiorum*. Fragment 1 is composed of upstream sequence of the target gene genomic DNA, and the first half of hygromycin-resistance gene (HY), and Fragment 2 is composed of the second half of hygromycin-resistance gene (YG) and downstream sequence of the target gene. Once transformed into protoplasts, the two fragments replace the target gene with wild-type genomic DNA by homologous recombination. (**B**) Color change in WT, 5–10, and *Sscle_10g079050* independent transgene complementation lines (Com-4 and Com-5) grown in PDB medium containing bromophenol blue indicator 48 hpi. (**C**) pH of WT, 5–10 and Com-4 and Com-5 lines after 48 hpi in PDB medium. (**D**) Lesions caused by *S. sclerotiorum* infection on lettuce leaves by the indicated fungal strains. Bar = 10 mm. Experiments were conducted at least three times with similar results. Data were recorded at 36 h post-inoculation (hpi). (**E**) Quantification of lesion size of WT, 5–10, Com-4 and Com-5 lines on lettuce leaves. ImageJ was used to analyze lesion size. Error bars represent SE. Statistical significance was analyzed using Student’s *t*-test. a, b and c represent different significance levels.

**Figure 4 ijms-23-11207-f004:**
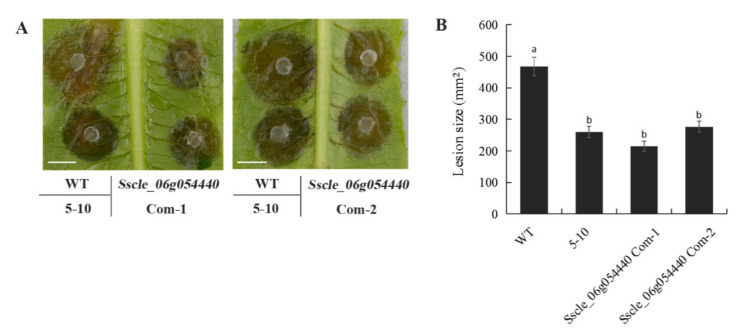
*Sscle_06g054440* failed to complement the defects of 5–10. (**A**) Necrotic lesions on lettuce leaf due to *S.sclerotiorum* infection by the indicated strains. Bar = 10 mm. Data were recorded at 36 h post-inoculation (hpi). (**B**) Quantification of the lesion size of WT, 5–10, *Sscle_06g054440* transgenic lines on lettuce leaves. ImageJ was used to analyze lesion size. Error bars represent SE. Statistical significance was analyzed using Student’s *t*-test. a and b represent different significance levels.

**Figure 5 ijms-23-11207-f005:**
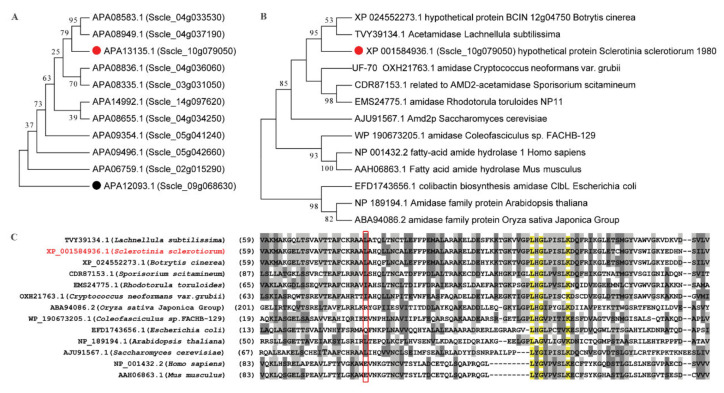
Sequence analysis of *Sscle_10g079050*. (**A**,**B**) The phylogenetic tree was generated by the ML method with bootstrap analysis (1000 bootstrap replicates) using an amino acid sequence alignment of homologous proteins within *S. sclerotiorum* (**A**) and among different species (**B**). APA12093.1 (*Sscle_09g068630*), annotated as a hydrolase, was selected as an outgroup. (**C**) Sequence alignment of homologous proteins from different species. Red box represents the homologous region of L82 in *Sscle_10g079050*. Numbers indicate the positions of the amino acids.

**Figure 6 ijms-23-11207-f006:**
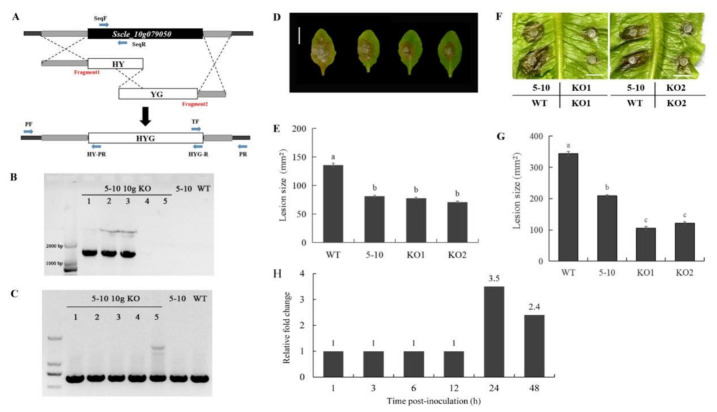
Knocking out *Sscle_10g079050* phenocopies 5–10 and exhibit reduced virulence. (**A**) Split marker method illustration. Fragment 1 is composed of the upstream sequence of the target gene and the first half of hygromycin-resistance gene (HY), and fragment 2 is composed of the second half of hygromycin-resistance gene (YG) and downstream sequence of the target gene. Once transformed into protoplasts, the two fragments can replace the target gene with hygromycin-resistance gene (HYG) by homologous recombination. (**B**) Agarose gel electrophoresis illustrating the fragments amplified with primers of PF and HY-PR. (**C**) Agarose gel electrophoresis illustrating the fragments amplified with primers of SeqF and SeqR. (**D**,**F**) Image of lesions on leaves of Arabidopsis (**D**) and lettuce (**F**) caused by fungal infection. Bar = 10 mm. Data were recorded at 36 h post-inoculation (hpi). (**E**,**G**) Quantification of lesion size of WT, 5–10, and *Sscle_10g079050* knockout mutants on leaves of *Arabidopsis* (**E**) and lettuce (**G**). ImageJ was used to analyze lesion size. Experiments were conducted at least three times. Error bars represent SE. Statistical significance was analyzed using Student’s *t*-Test. a, b in (**E**) and a, b, c in (**G**) represent different significance levels. (**H**) Expression of *Sscle_10g079050* during infection of *Brassica napus*. Data were extracted from publicly available RNA-Seq dataset [27]. Fold change relative to 0 h post inoculation (hpi). 1: No significant change in expression.

## Data Availability

Not applicable.

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
