# Peer review of "An Amidase Contributes to Full Virulence of *Sclerotinia sclerotiorum"

_ijms, 2022, doi:10.3390/ijms231911207_

Round 1
Reviewer 1 Report
A mutant with weak pathogenicity was obtained from the wild-type strain of Sclerotinia sclerotiorum by UV mutagenesis. Through genome sequencing analysis and gene knockout and complementation tests, it was identified that the mutation was related to the gene encoding amidase (Sscle_06g054440). The content of oxalic acid in the culture medium of the mutant was determined and found to be significantly decreased. Because oxalic acid is an important pathogenic factor of S. sclerotiorum, the authors concluded that amidase affects pathogenicity by regulating the level of oxalic acid. This study has important significance for further understanding the pathogenic mechanism of S. sclerotiorum.
However, the reviewer thinks that the reduction of oxalic acid level caused by the destruction of Sscle_06g054440 may be indirect, and the accumulation of oxalic acid is related to both the synthesis and degradation. The authors have not further clarified the mechanism of amidase regulating the synthesis or degradation of oxalic acid, and have not even detected the expression of 06g054440 in the pathogenic process of S. sclerotiorum and the growth on PDA medium. There are many factors that can affect the pathogenicity of S. sclerotiorum, including some plant cell wall degrading enzymes and effectors, why did authors only focused on oxalic acid? The reviewer thinks that the study is still at a very preliminary stage, and suggests that the authors compare the similarities and differences of transcriptome among Sscle_06g054440-knockout transformants, UV-mutants and wild-type strains in the process of invading plants and growing on plants, clarify the metabolic and signaling pathways related to the impact of amidase on the pathogenicity of S. sclerotiorum, and find the relationship between amidase and oxalic acid accumulation and other pathogenic factors. Because this enzyme is likely to be related to metabolism, if possible, the reviewer suggests analyzing the metabolome of mutants and wild strains of S. sclerotiorum.
Reviewer 2 Report
Dear authors
I reviewed the manuscript entitled “An amidase contributes to the virulence of Sclerotinia sclerotiorum through regulating oxalic acid levels”, which was performed using good methodologies. Authors found a gene codifying for one amidase which is involved in the virulence of Sclerotinia sclerotiorum via oxalic acid.
I have only minor suggestions
For all the manuscript, the unities must be separated from numerical values. In example 12 h (lines 76, 255, 320).
In my opinion, the title of figures must be increased in quality.
Round 2
Reviewer 1 Report
The authors added some new evidence to demonstrate that amidase gene of S. sclerotiorum is up-regulated in the process of plant infection, which is very important. Because the authors are still unable to provide data on how to affect oxalic acid synthesis or degradation, and the authors are also unable to provide evidence on whether amidase affects other pathogenic factors (such as PCWDEs and effectors). It may be true that amidase directly regulates oxalic acid accumulation, but there is no evidence, it still is a presumption. Therefore, In order to make the paper more rigorous, I do not agree to highlight the effect of amidase on oxalic acid accumulation and thus affect pathogenicity in the title. The authors may know that S. sclerotiorum also has the ability to degrade oxalic acid, and the data in PDA medium or PDB medium may only be an overall result. Therefore, I suggest that the authors change the title to "An amidase contributions to full virulence of Sclerotinia sclerotiorum" and discuss amidase may be involved in the accumulation of oxalic acid. Replacing the original title with this one, this research is still of great significance.
Author Response
We thank the reviewer for your quick review of our revised manuscript. We totally agree with your criticism. We have now changed the title according to your great suggestion. In addition, we added in discussion that "The amidase may even affect OA-independent processes involved in virulence" to highlight that the amidase may not be restricted to affecting OA levels.
Round 3
Reviewer 1 Report
I do not have any more comments on this manuscript.
Author Response
solved